

# A mathematical approach to understanding emergent constraints

Femke J. M. M. Nijsse[1,2] and Henk A. Dijkstra[1,3]

[1]Institute for Marine and Atmospheric research Utrecht, Department of Physics, Utrecht University, Utrecht, the Netherlands
[2]College of Engineering, Mathematics and Physical Science, University of Exeter, UK
[3]Center for Complex Systems Science, Utrecht University, Utrecht, the Netherlands.

*Correspondence to:* Femke Nijsse <fn235@exeter.ac.uk>

**Abstract.** One of the approaches to constrain uncertainty in climate models is the identification of emergent constraints. These are physically explainable empirical relationships between a particular simulated characteristic of the current climate versus future climate change from an ensemble of climate models, which can be exploited using current observations. In this paper, we develop a theory to understand the appearance of such emergent constraints. Based on this theory, we also propose a classification for emergent constraints and applications are shown for several idealized climate models.

## 1 Introduction

Improving the accuracy of climate projections is one of the most important challenges in climate modeling. The uncertainty can be reduced by the development of more and more sophisticated global climate models, capturing more processes and scales. However, the societal importance of climate projections calls for a faster pace of improvement and alternative approaches that aim to better determine the accuracy of existing models. One of the proposed methods to accomplish this has been the use of so-called emergent constraints, where current observations are used to constrain future projections (Collins et al., 2012).

In multimodel ensembles of complex climate models, an apparent linear relation can be found between short-term and long-term changing variables. More credibility is attached to models that match the observed variability or trend well over the recent period. In this way, current observations provide a constraint to long term trends. The observable variable is called the predictor, while the variable that is to be constrained is called the predictand (Klein and Hall, 2015). In recent years, emergent constraints have been found for Arctic warming, snow albedo feedback, tropical carbon, the global precipitation among other variables (Allen and Ingram, 2002; Bracegirdle and Stephenson, 2013; Hall and Qu, 2006; Wenzel et al., 2014) and more recently, climate sensitivity (Cox et al., 2018).

A prominent example is the emergent constraint found in Hall and Qu (2006) where an emergent relationship was found between the strength of the snow-albedo feedback (SAF) on a seasonal time scale and the SAF under global warming in a CMIP4 ensemble. They also elucidated the key physical process behind the emergent constraint. Models where the maximum albedo of snow is highest have the largest SAF on both time scales because the contrast between snow-covered and snow-free areas is high (Qu and Hall, 2007).

However, a more general dynamical picture on how emergent constraints occur in multi-model ensembles or even in a parameter ensemble of a single model is still lacking. Under which circumstances are these constraints expected to arise?





Some emergent constraints may be spurious and could arise because of shared errors in a particular multimodel ensemble (Bracegirdle and Stephenson, 2013). A mathematical framework is desired to identify spurious constraints and to give an indication as to where new emergent constraints might arise.

Here, we investigate how and under what conditions emergent constraints appear and what can be learned about the physics
of the climate system. We will use Linear Response Theory (LRT) to address the problem of forcing-response relations on different time-scales (Risken, 1996). Ruelle demonstrated that LRT can be extended to study the response of non-equilibrium systems to external forcing. As with the fluctuation-dissipation theory, Ruelle's LRT uses the statistical properties of the unforced (equilibrium) state only but it does not assume (quasi)-equilibrium. Recently, LRT has been proposed as a rigorous framework for computing the response of the climate system and its applicability has been tested on the Lorenz-96 model and
on the idealized global climate model PlaSim (Lucarini and Sarno, 2011; Ragone et al., 2016).

The paper is organized as follows. To obtain an understanding of emergent constraints we start by formulating a mathematical framework in terms of susceptibilities by making use of LRT (section 2). This results in explicit expressions for the appearance of emergent constraints in terms of susceptibility functions. In section 3 a classification scheme for emergent constraints is proposed. Then, in section 4, applications are presented for conceptual climate models, such as Ornstein-Uhlenbeck processes
in one and two dimensions, an energy balance model and the PlaSim model. The results are summarized and discussed in section 5.

## 2   Response functions

In this section explicit expressions are given for response functions of the state of a dynamical system which depends on a single parameter and which is subjected to a non-stationary forcing. Such response functions are used in the following section
to classify the different emergent constraints.

We illustrate the approach using the general one-dimensional forced Stochastic Differential Equation (SDE)

$$dX_t = (-V'(X_t) + F(t))dt + \sqrt{\sigma}dW_t. \tag{2.1}$$

Here $V(x)$ is a smooth confining potential, meaning that a equilibrium solution exists for the unforced system (Pavliotis, 2014), and $F(t)$ is a prescribed forcing. Furthermore, $\sigma$ is the noise amplitude and the associated Wiener process is indicated by $W_t$.
Usually, the potential depends on a parameter. For example, when $V'(x) = \gamma x$, the solution of the unforced problem is the well-known Ornstein-Uhlenbeck (OU) process.

The probability density function of the unforced ($F(t) = 0$) system, say $\bar{p}$, satisfies the Fokker-Planck equation

$$\frac{\partial \bar{p}}{\partial t} = \frac{\partial (V'(x)\bar{p})}{\partial x} + \frac{\sigma}{2}\frac{\partial^2 \bar{p}}{\partial x^2} = \mathscr{L}^*(\bar{p}) \tag{2.2}$$

which defines the Fokker-Planck operator $\mathscr{L}^*$. The equilibrium distribution of the unforced system, here indicated by $\bar{p}_e$, is
given by

$$\bar{p}_e(x) = \frac{1}{Z}e^{\frac{-2V(x)}{\sigma}}; \qquad Z = \int\limits_{-\infty}^{\infty} e^{\frac{-2V(x)}{\sigma}} \, dx. \tag{2.3}$$



Linear response theory (Ragone et al., 2016) provides an expression for the change in the expectation value of the change in an observable $O$ (e.g. the temperature, ice extent or the standard deviation of either), say $\Delta O(t)$ when the system is forced, compared to the unforced case, i.e.

$$\Delta O(t) = E[O(X_t)] - E[O_e(X_t)], \tag{2.4}$$

where again the subscript $e$ indicates the equilibrium of the unforced system. It follows that

$$\Delta O(t) = \int_0^t R_O(t-s)F(s)ds; \qquad R_O(t) = \int_{-\infty}^{\infty} O(x)\, e^{\mathscr{L}^* t}(-\frac{\partial \bar{p}_e}{\partial x})dx, \tag{2.5}$$

where $R_O(t)$ is the response function, which is extended to be zero for $t < 0$ to ensure causality. When (2.5) is Fourier transformed we find, using the convolution theorem,

$$\mathcal{F}(\Delta O(t))(\omega) = \chi(\omega)\hat{F}(\omega), \tag{2.6}$$

where the Fourier transform $\chi(\omega)$ of the response function $R_O(t)$ is the susceptibility. If we take a cosine forcing, i.e. $F(t) = F_0 \cos \omega_0 t$ then $\hat{F}(\omega) = F_0 \pi(\delta(\omega - \omega_0) + \delta(\omega + \omega_0))$ so once we know $\chi(\omega)$, we can determine the response $\Delta O(t)$.

In the appendix, it is shown that taking the mean value as our observable, i.e. for $O = x$, the response function and its corresponding susceptibility can be written as

$$R_O(t) = \frac{2}{\sigma}\sum_{l=1}^{\infty}\beta_l e^{-\lambda_l t}, \qquad \chi(\omega) = \frac{2}{\sigma}\sum_{l=1}^{\infty}\frac{\beta_l}{\lambda_l + i\omega} \tag{2.7}$$

where $\lambda_l$ are the eigenvalues of the so-called generator $\mathscr{L}$

$$\mathscr{L}u = V'(x)\frac{\partial u}{\partial x} + \frac{\sigma}{2}\frac{\partial^2 u}{\partial x^2}, \tag{2.8}$$

and the $\beta_l$ are projection coefficients that indicate how strongly the system responds to the forcing See the appendix for a more detailed description. For the OU process, these eigenvalues and eigenfunctions are given by (Pavliotis, 2014)

$$\lambda_l = \gamma l\,; \qquad \phi_l(x) = \frac{1}{\sqrt{l!}}H_n(\sqrt{\frac{2\gamma}{\sigma}}x), \tag{2.9}$$

where $H_n$ are the Hermite polynomials.

Going back to the general case, the amplitude $A$ of the response to a periodic forcing $F(t) = F_0 \cos \omega_0 t$ is determined by the absolute value of the susceptibility

$$A(\Delta X(t))(\omega_0) = \frac{2F_0}{\sigma}\sum_{l=1}^{\infty}\frac{\beta_l}{\sqrt{\lambda_l^2 + \omega_0^2}}. \tag{2.10}$$

The previous analysis can be generalised to more dimensions. In two dimensions, for example, with a vector $Y_t = (Y_{1t}, Y_{2t})^T$,

the SDE becomes

$$dY_t = (-\nabla V(Y_t) + F(t)\hat{i})dt + \sqrt{\sigma}I_2 dW_t, \tag{2.11}$$



where the term $F(t)\hat{i}$ denotes a forcing in the direction of the first variable and $I_2$ the identity matrix. As shown in Pavliotis (2014), the derivation of the response function follows the one-dimensional case closely, resulting in:

$$R_{Y_1}(t) = \frac{2}{\sigma} \sum_{l=1}^{\infty} g_l e^{-\lambda_l t}; \qquad\qquad R_{Y_2}(t) = \frac{2}{\sigma} \sum_{l=1}^{\infty} h_l e^{-\lambda_l t}, \qquad (2.12)$$

where $g_l$ and $h_l$ are again projection coefficients, $g_l$ and $h_l$ containing a term describing strength of the response in $Y_1$ and $Y_2$ respectively. Calculating the response is analogous to the one-dimensional case, so that the Fourier transforms of the response functions are given by

$$A\left(\Delta Y_1(t)\right)(\omega_0) = \frac{2}{\sigma} \sum_{l=1}^{\infty} \frac{g_l}{\sqrt{\lambda_l^2 + \omega_0^2}}; \qquad\qquad A(\Delta Y_2(t))(\omega_0) = \frac{2}{\sigma} \sum_{l=1}^{\infty} \frac{h_l}{\sqrt{\lambda_l^2 + \omega_0^2}} \qquad (2.13)$$

Note that generalizing to uncoupled multidimensional systems, the eigenfunctions are found to be the tensor products of the eigenfunctions in the one-dimensional case, while the corresponding eigenvalues are the sum of the eigenvalues in the one-dimensional case. For an Ornstein-Uhlenbeck process the eigenfunctions $\phi_{l,m}$ are given by

$$\phi_{l,m} = \phi_l(y_1)\phi_m(y_2),$$

for $\phi_l$ as defined in (2.9); these also form an orthonormal complete basis.

## 3 Classification of emergent constraints

Although a wide set of different emergent constraints have been found, no attempts have been made to classify them so far using dynamical criteria. Here, a classification is proposed based on the time-characteristics of the predictor and on the relationship between the predictor and the predictand. Using this classification, assessment of their applicability becomes easier. Furthermore, a classification is a prerequisite for a dynamical description of emergent constraints.

Firstly, an emergent constraint can be either direct or indirect. In the direct case, the predictor and predictand are the same observable, while in the indirect case the predictor variable and predictand variable have to be linked via a physical process. We make a further distinction between static and dynamic emergent constraints. In a dynamic emergent constraint a response to a known, or sometimes even unknown, forcing in the (present-day) predictor is linked to the response of the (future) predictand under the same (or a similar) forcing. For example: the forcing can be the annual cycle of solar radiation, but can also be caused by ENSO or historical climate change. In a static emergent constraint a relationship between the time-independent quantity of the unforced system in the present-day (predictor) is linked to the response in a quantity under climate change.

As an illustration, we apply our classification to examples of emergent constraints found in the literature in Table 1. Although this is not a complete overview, examples are found of the four types of emergent constraints. There are many examples of direct dynamical constraints, such as the one involving the snow-albedo feedback shown in figure 1 (Hall and Qu, 2006). Dynamic direct emergent constraints are the most intuitive. As long as the variations in the predictor are of a sufficient amplitude compared to those of the predictand, a correlation between the predictor and predictand automatically points towards a common





physical basis, for example a common dynamical response to an external forcing. The direct static emergent constraint found by (Bracegirdle and Stephenson, 2013) makes use of spatial patterns. All of the indirect constraints involve equilibrium climate sensitivity as the predictand. Often a bias in the model ensemble is linked to ECS. For instance, in Tian (2015) the asymmetry bias in ITCZ is linked to climate sensitivity. An example of a dynamical indirect emergent constraint is provided by Cox et al.

(2018), who relate a function of autocorrelation of global surface temperature to ECS.

Based on the response function theory in section 2, we further elaborate on the classification and also discuss conditions for each type of constraint for a dynamical system with varying parameters (which defines the ensemble of models).

For a direct dynamical emergent constraint, the ratio of the responses to the frequencies $\omega_1$ and $\omega_2$ should be constant over the (parameter) ensemble members $e_i$ (for simplicity, we only consider linear relationships). For the simple case of two forcings

that only differ in frequency, we find the condition from the ratio of the susceptibilities SR as

$$\text{SR}(e) = \frac{A(\Delta O(t))(\omega_2)}{A(\Delta O(t))(\omega_1)} = \frac{\sum_{l=1}^{\infty} \frac{\beta_l}{\sqrt{\lambda_l^2 + \omega_2^2}}}{\sum_{l=1}^{\infty} \frac{\beta_l}{\sqrt{\lambda_l^2 + \omega_1^2}}} = C^{st}, \tag{3.1}$$

where $C^{st}$ indicates a quantity independent of the parameter generating the ensemble. Often we are interested in the impact of one variable ($B$) on the other ($O$) under external forcings $F_1$ and $F_2$ that differ slightly. The condition has to be adjusted to:

$$\text{SR}(e) = \frac{A(\Delta O(t)|_{F_2})(\omega_2)}{A(\Delta O(t)|_{F_1})(\omega_1)} \frac{A(\Delta B(t)|_{F_1})(\omega_1)}{A(\Delta B(t)|_{F_2})(\omega_2)} = C^{st}, \tag{3.2}$$

This is further discussed in the example of the idealized energy balance model. For the Ornstein-Uhlenbeck case, the ratio of response amplitudes reduces to

$$SR(\gamma) = \frac{\beta_1/\sqrt{\lambda_1^2 + \omega_2^2}}{\beta_1/\sqrt{\lambda_1^2 + \omega_1^2}} = \frac{\sqrt{1 + \left(\frac{\omega_1}{\gamma}\right)^2}}{\sqrt{1 + \left(\frac{\omega_2}{\gamma}\right)^2}}, \tag{3.3}$$

since both the observable and the derivative of the potential are orthogonal to all eigenfunctions other than $\phi_1$. This ratio is dependent on $\gamma$. In the case $\gamma \gg \omega_i$ for $i \in \{1, 2\}$ this ratio is nearly one and an emergent relationship is present for a model

ensemble generated by varying $\gamma$.

Physically, we expect that the same mechanisms to be responsible for the response at a short and long time scale to obtain this type of emergent constraint. The system should have response times smaller than the time scale of the forcing or equivalently: the generator should have eigenvalues $\lambda$ larger than the frequency of the forcing. Naturally, the response times $\frac{1}{\lambda}$ of the dominant processes are expected to be at least smaller than the time scale of the slow forcing $\frac{1}{\omega_1}$.

Mathematically, the ratio in (3.1) becomes one in the case that all eigenvalues $\lambda_l$ are much larger than the forcing frequencies. Interestingly, the linear relation breaks down in the case that the fast forcing has the same order of magnitude as the eigenvalues of the dominant terms in the susceptibility. Under the assumption of a single dominant term in the susceptibility and a slow forcing with frequency $\omega_2 \to 0$ the first correction term to the slope-one linear relation between predictand $y$ and predictor $x$, is cubic in $x$.





In the case of indirect dynamical emergent constraints, a relationship between a predictor $Y_1$ and a predictand $Y_2$ is found. Assuming the predictor $Y_1$ is again a response to some forcing, we can repeat the analysis above for direct constraints for a system of two dimensions, where a forcing is added in one direction. Mutatis mutandis, a condition very similar to (3.1) is found, as

$$\frac{A(\Delta Y_1(t))(\omega_2)}{A(\Delta Y_2(t))(\omega_1)} = \frac{\sum_{l=1}^{\infty} \frac{g_l}{\sqrt{\lambda_l^2 + \omega_2^2}}}{\sum_{l=1}^{\infty} \frac{h_l}{\sqrt{\lambda_l^2 + \omega_1^2}}} = C^{st}, \tag{3.4}$$

where $g_l$ and $h_l$ are defined as in (2.12). For an emerging constraint to exist, the projection terms of the different observables should thus change in a similar fashion under the change in parameter.

Static direct constraints link the mean of an observable to a change in the system under a specific forcing. Note that the susceptibility only contains information about the response to such forcing. Even in the limit of $\omega \to 0$, it denotes the linear response of the system, without any information on the mean state (Lucarini and Sarno, 2011). So, to derive the condition for a linear relationship the mean $E[O_e(X_t)] = \int_{-\infty}^{\infty} \bar{p}_e O(x)\, dx$ and the susceptibility at frequency $\omega_1$ are used.

For static emergent constraints, the linear relationship between the response and the mean state is not expected to pass through the origin, since the mean will in general be nonzero. Therefore, an additional term $I$ is added to the ratio, denoting the intercept of the line between the mean state and the response with varying parameters. Instead, the susceptibility is compared to the mean state and the following condition is derived, where $C^{st}$ should again be a constant independent of parameter(s) that is used to generate the ensemble:

$$\frac{E[O_{1t}] - I}{A(\Delta O_2(t))(\omega_1)} = \frac{\int_{-\infty}^{\infty} \bar{p}_e O_1(x)\, dx - I}{\sum_{l=1}^{\infty} \frac{2}{\sigma} \frac{h_l}{\sqrt{\lambda_l^2 + \omega_1^2}}} = C^{st}. \tag{3.5}$$

Again $C^{st}$ can either be positive or negative, depending on the physics under consideration. This equation is both valid for direct and indirect static emergent relationships; the term $h_l$ contains either the information about the same observable of the predictor (direct) or the information of a different variable than the predictor (indirect).

## 4 Application to idealized climate models

From the previous sections, it appears that the computation of the eigensolutions of the generator of the dynamical system are central to determine whether an emergent constraint will appear or not. In this section, we will provide examples using idealized climate models.

The eigenvalues and eigenfunctions of the generator were numerically determined using the fact that the eigenvalues of the Fokker-Planck operator $\mathscr{L}^*$ are equal to those of the generator and that the eigenfunctions can be computed from the transformation: $\phi_l = \phi_l^*/\bar{p}_e$. The Fokker-Planck operator was discretized with use of Chang-Cooper algorithm (Chang and Cooper, 1970). Eigenvalues and eigenvectors were determined using an Implicitly Restarted Arnoldi Method (Lehoucq et al., 1998). Explicit simulations of the SDEs were performed the using a stochastic Runge-Kutta method (Kloeden and Platen, 1992).




## 4.1 Ornstein-Uhlenbeck cases

First, the one-dimensional Ornstein-Uhlenbeck process is considered with SDE

$$dX_t = (-\gamma X_t + F_i(t))dt + \sqrt{\sigma}dW_t, \tag{4.1}$$

forcing $F_i(t) = \sin 2\pi t\omega_i$ and frequencies $\omega_1 = 0.001$ and $\omega_2 = 0.1$. A parameter ensemble is created by varying $\gamma$. In this case,
analytic solutions exist for the eigenvalues and eigenvectors of the generator. Eigenvectors and eigenvalues were determined
using the Chang-Cooper scheme on a domain $[-25, 25]$ with $\Delta x = 0.25$. The numerically computed susceptibilities, as shown
in figure 2b, are in agreement with the analytic ones and capture the response (figure 2a) well, as expected in this linear case.

In the two-dimensional Ornstein-Uhlenbeck case, the same forcing $F_i(t)$ is added but only in the first dimension. The
governing SDE is given by

$$dX_t = \left[ \begin{pmatrix} -\gamma_1 & \delta \\ \delta & -\gamma_2 \end{pmatrix} X_t + \begin{pmatrix} F_i(t) \\ 0 \end{pmatrix} \right] dt + \sqrt{\sigma} \begin{pmatrix} 1 & 0 \\ 0 & 1 \end{pmatrix} dW_t, \tag{4.2}$$

and a parameter ensemble is generated by changing the damping rate $\gamma_1$. Two ensembles are compared with $\delta = 0.2$ in the first
ensemble and $\delta = 0.5$ in the second. The damping term $\gamma_2$ is held constant at $\gamma_2 = 0.7$.

In figure 3, the eigenvalues and susceptibility ratios are plotted. In the case of a relatively weak coupling ($\delta = 0.2$) all nonzero
eigenvalues are larger than the fast forcing frequency $\omega_2$, so the system response time is smaller than the forcing time scales.
On the other hand, the strong coupling ($\delta = 0.5$) leads to a slow down of the system, so that some eigenvalues now become
smaller than $\omega_2$. In these cases ($\gamma_1 < 0.5$) the system does not have time to portray the full response to a forcing, while for
others ($\gamma_1 > 0.5$) it does. Consequently, the strength of the response actually decreases for $\gamma_1 < 0.5$. Directly calculating the
expectation value as the mean of 500 stochastic trajectories confirms this result (not shown).

The results in figure 4 show a large variation over the ensemble in the projection term of the predictor on the eigenfunctions
($g_l$, see appendix). In constrast, the product of the two projection terms in the predictand ($h_l$) changes relatively little over
the ensemble for both coupling strengths. Even though the projection terms now play a significant role in determining the
response, the eigenvalues still determine whether the relation is linear (fast compared to forcing) or nonlinear (similar size to
forcing frequency). In the weak-coupling system, the susceptibility ratio is almost constant and an emergent linear relationship
is found. The strong-coupling system does only portray an emergent relationship for certain regimes (low or high $\gamma_1$). A case
can be made though that the highly coupled system is the system for which finding an emergent constraint is more likely,
because the strength of the response is substantially higher.

## 4.2 Energy Balance model

In this section a specific emergent constraint is examined in more detail, namely the one pertaining to the snow-albedo feedback
(SAF) first described by (Hall and Qu, 2006). They found a correlation between SAF on a seasonal scale and SAF as a result
of climate change. In models with a high snow albedo, the contrast between snow-covered and bare surfaces was largest and
consequently the sensitivity to changes in temperature was largest (Qu and Hall, 2007). To study this emergent constraint we





modify a simple energy balance model and make the albedo temperature-dependent. A parameter in the albedo function will be used to define a parameter ensemble.

With constant albedo, the energy balance model reads:

$$dT = \frac{1}{c_T}\left(Q(1-\alpha) + A\ln\frac{C}{C_{ref}} + G - \epsilon\sigma_B T^4\right)dt + \sqrt{\sigma_T}\,dW_t, \tag{4.3}$$

where $dT$ is the temperature change, $c_T$ the atmospheric heat capacity, $Q$ the solar insolation, $\alpha$ the albedo, $C$ the concentration of greenhouse gases, $C_{ref}$ a reference concentration, $G$ represents the radiative forcing due to the reference greenhouse gas concentration, $\sigma_B$ the Stephan-Boltzmann constant and $\epsilon$ the emissivity of the Earth. The standard parameter values for this model can be found in Table 2. The parameters of the albedo function are chosen to ensure that no bistability is present in the model, in which case LRT would break down.

Before examining the snow-albedo feedback, note that for some variables, notably the climate sensitivity, a simple EBM can react differently to forcing from solar insolation or greenhouse gases. This can be determined from, with $H = G + A\ln\frac{C}{C_{ref}}$ and for a value of $\epsilon = 1$,

$$\frac{\partial}{\partial\alpha}\frac{\partial T}{\partial Q} = \frac{\sigma^{1/4}}{4\left(Q(1-\alpha)+H\right)^{3/4}}\left(\frac{3Q(1-\alpha)}{Q(1-\alpha)+H} - 1\right) < 0; \qquad \frac{\partial}{\partial\alpha}\frac{\partial T}{\partial H} = \frac{3Q\sigma^{1/4}}{16\left(Q(1-\alpha)+H\right)^{7/4}} > 0 \tag{4.4}$$

Sensitivity to greenhouse forcing decreases when albedo decreases, while sensitivity to solar insolation (seasonal sensitivity)
increases for an increasing albedo, using typical values for $Q$ and $H$.

To mimic the physical mechanism behind the emergent constraint, the albedo is taken to be temperature dependent, i.e., for low (high) temperatures, the albedo is high (low). A logistic function is used to model this effect,

$$\alpha(T) = \alpha_{min} + \frac{\alpha_{amp}}{1 + \exp\left(k(T - T_h)\right)} \tag{4.5}$$

where $\alpha_{min}$ is the minimum albedo, $\alpha_{amp}$ is the amplitude, $k$ is a steepness factor and $T_h$ is the temperature at which half of
20 the amplitude is reached. The amplitude $\alpha_{amp}$ is the parameter that is varied over the ensemble.

In the first case, the insolation forcing is given by $Q = Q_0(1 + Q_s\sin 2\pi t/\tau)$ where $\tau$ corresponds to one year and $Q_s$ is a seasonal modulation amplitude, with parameter values are shown in Table 2. The snow-albedo feedback term is then computed by dividing the amplitude of the albedo cycle by the amplitude of the temperature cycle. A second case is considered in which the greenhouse gas concentration $C$ is increased 0.3% per year from 295 ppmv over a period of 300 year. Here the snow-albedo
feedback is computed by dividing the total albedo response by the total temperature response. In each case, the variance of the noise $\sigma_T$ in (4.3) was chosen as $10^{-7}$ K$^2$/s. Changing this parameter does not influence the eigenvalues as expected from the theory (Pavliotis, 2014). While the projections of the eigenvalues and eigenfunctions did change slightly, the susceptibility ratio was not influenced significantly by a variation of the $\sigma_T$ (halving and doubling of $\sigma$, not shown). In the computation of the solution of the Fokker-Planck equation using the Chang-Cooper scheme, we used a resolution of $1$ K which is sufficient to
accurately determine the eigenvalues and eigenfunctions of the generator.

As mentioned above, application of equation 3.1 is not self-evident. Considering temperature to be a forcing ignores the fact that temperature responds differently to seasonal and greenhouse gas forcing, as shown in equation 4.4. Secondly, using $d\alpha/dT$



as the observable directly does not work either. Linear response theory does not give the expectation value of the observable, but the expectation value of the deviation due to the forcing, while we are interested in the change due to a parameter change.

Instead, the SAF can be described by two observables: SAF is determined by taking the ratio of the susceptibilities of albedo to temperature. Therefore, we use the modified equation (3.2):

$$\text{RFS}(\alpha_{amp}) = \frac{A(\Delta\alpha(t)|_Q)(\omega_2)}{A(\Delta\alpha(t)|_C)(\omega_1)} \frac{A(\Delta T(t)|_C)(\omega_1)}{A(\Delta T(t)|_Q)(\omega_2)} = \frac{\sum_{l=1}^{\infty} \frac{\alpha_l}{\sqrt{\lambda_l^2 + \omega_2^2}}}{\sum_{l=1}^{\infty} \frac{\gamma_l}{\sqrt{\lambda_l^2 + \omega_1^2}}} \frac{\sum_{l=1}^{\infty} \frac{\delta_l}{\sqrt{\lambda_l^2 + \omega_2^2}}}{\sum_{l=1}^{\infty} \frac{\beta_l}{\sqrt{\lambda_l^2 + \omega_2^2}}} = C^{st},$$

(4.6)

where

$$\alpha_l = \langle \alpha, \phi_l \rangle_{\bar{p}_e} \langle (1 - \alpha(T)) V'(T), \phi_l \rangle_{\bar{p}_e}, \qquad \gamma_l = \langle \alpha, \phi_l \rangle_{\bar{p}_e} \langle V'(T), \phi_l \rangle_{\bar{p}_e}$$

$$\beta_l = \langle T, \phi_l \rangle_{\bar{p}_e} \langle (1 - \alpha(T)) V'(T), \phi_l \rangle_{\bar{p}_e}, \qquad \delta_l = \langle T, \phi_l \rangle_{\bar{p}_e} \langle V'(T), \phi_l \rangle_{\bar{p}_e}. \qquad (4.7)$$

In the case the susceptibilities are all dominated by one term with index $l$, this reduces to $C^{st} = (\alpha_l \delta_l)/(\beta_l \gamma_l) = 1$

In figure 5 the sensitivity of temperature to varying amplitude of the albedo function is shown, as well as the sensitivity
of the snow-albedo feedback and condition for the existence of an emergent constraint. As shown in figure 5a, no emergent relationship is found for climate sensitivity, a feature that was analytically found in the case of constant albedo. In figure 5b the emergent constraint on SAF is shown. In the warm regime (low albedo, lower line in the figure), the SAF becomes larger for larger $\alpha_{amp}$. The larger the maximum albedo, the steeper the logistic albedo function. A second effect also takes place: with higher maximum albedo, the warmer it gets. Consequently, sensitivity of the albedo function is smaller. This decrease
in sensitivity also takes place in the cold regime; the colder it gets, the less sensitive the albedo gets. In the cold regime it is clear that this second mechanism dominates. The results can be reproduced by use of LRT, as shown in figure 5c and 5d. The discrepancies disappear when forcing is small; the climate change forcing in particular is causing most of the differences.

One can extend the energy balance model by representing the response of snow and ice explicitly as a relaxation towards the logistic reference albedo function $\alpha(T)$ given in (4.5). This gives the extended model

$$20 \quad dT = \frac{1}{c_T} \left( Q(1 - \bar{\alpha}) + H - \epsilon \sigma T^4 \right) dt + \sqrt{\sigma_T}\, dW_t \qquad d\bar{\alpha} = -\frac{1}{\tau_s} \left( \bar{\alpha} - \alpha(T) \right) dt + \sqrt{\sigma_{\bar{\alpha}}}\, dW_t, \qquad (4.8)$$

where $\tau_s = 4 \times 10^6$s is the response time of the albedo. The drift term in the Fokker-Planck equation corresponding to (4.8) is not the gradient of a potential but the eigensolutions of the generator can of course still be computed numerically.

Extending the model with an explicit albedo function does not change the dynamics of the system significantly, nor the eigenvalues and eigenvectors. Figure 6b shows the eigenvalues of the extended EBM to be almost exactly equal to the eigen-
25 values of the original model, the imaginary parts continuing to be zero. The projection coefficients are very similar as well (not shown). Thus, the inclusion of a smaller time scale does not improve the response.

## 4.3 PlaSim

To bridge the gap between parameter ensembles in simple dynamical systems and Earth System Models, the SAF emergent constraint is further examined in PlaSim. PlaSim is a numerical model of intermediate complexity, developed at the University



of Hamburg to provide a fairly realistic present climate which can still be simulated on a personal computer (Fraedrich et al., 2005). The atmospheric dynamics are modelled using the primitive equations formulated for temperature, vorticity, divergence and surface pressure. Moisture is included by transport of water vapor. The equations are solved using the spectral method. A full set of parameterizations is used for unresolved processed such as long and shortwave radiation with interactive clouds, boundary layer fluxes of latent and sensible heat and diffusion.

In this climate model snow albedo is a function of surface temperature $T_s$, snow depth and vegetation cover. The bare soil snow albedo in PlaSim is described by:

$$A_{snow} = \begin{cases} A_{max}, & \text{if } T_s \leq 10^\circ \text{ C.} \\ A_{min}, & \text{if } T_s > 0^\circ \text{ C.} \\ A_{min} - (A_{max} - A_{min})\frac{T_s}{10} & \text{otherwise.} \end{cases} \quad (4.9)$$

This equation is modified in the presence of vegetation and in the case of shallow snow depth. See Lunkeit et al. (2011) for more details. A set of simulations was performed with $A_{max}$ varying between 0.650 and 0.900. The historical forcing in PlaSim was approximated by a $CO_2$ increase from 295 ppm at a rate of 0.3% per year in the 20th century and 1% per year in the 21st century before it stabilised at 720 ppm; a 50-year spin-up was used.

In figure 7 the PlaSim results are shown which can be compared to the results from Hall and Qu (2006) in figure 1. Note that the variation in CMIP4 is significantly larger than the variation found in PlaSim, but that the PlaSim results fit on the relation found by Hall and Qu (2006). Variations in other parameterizations, such as the maximum snow albedo over forested regions, increase the spread in PlaSim SAF further (not shown). This simulation shows that the constraint that emerges in a multi-model ensemble with structurally different formulations of the snow response can to some extent also be reproduced using variations in one parameter. This provides the justification for simplifying further to energy balance models to examine the SAF emergent constraint.

## 5   Summary, Discussion and Conclusions

In this paper, we have presented a dynamical framework behind the occurrence of emergent constraints in parameter dependent stochastic dynamical systems. In these systems, emergent constraints are related to ratios of response functions which can be determined using linear response theory. It was shown that for a large class of systems, these ratios could be expressed in terms of eigenvalues and projections on eigenvectors of the generator of the system.

A classification of emergent constraints was given and several types could be distinguished depending on whether similar (direct) or different (indirect) observables are considered and whether a response in present-day climate (dynamical) or the time-independent part of present-day climate (static) is linked to a response of the future climate system. For a linear dynamical emergent constraint, the ratio of susceptibilities at the two frequencies under consideration should be a positive constant over the ensemble. When the response is computed with respect to an internal variable (in contrast to an external forcing), a condition is



posed on the susceptibilities of the two observables in the system. Static constraints are encountered when a linear relationship is found between the expectation value of the observable and the susceptibility at the frequency of the forcing.

Examples were given using several idealized climate models and in particular the emergent constraints involving the snow-albedo feedback was considered in detail. We found that dynamical emergent relationships can occur when the time scale of

the system, indicated by the eigenvalues, changes with the parameter and is smaller than the forcing time scales. When this condition is not met, deviations from linearity occur. When the linearity of the relation is exploited in further analysis, such as in the interpretation of emergent constraints by Wenzel et al. (2014), this might lead to a bias in the estimate of the predictand. This is of particular interest because differences in response size between climate models is often determined by feedbacks strength in climate systems. Larger feedbacks give rise to larger timescales (Roe, 2009), which is reflected in the eigenvalues

of the generator. For an emergent constraint on a feedback quantity a more complicated constraint mechanism occurs, where one has to take into account the response to two different observables, which typically have different time scales.

The classification of emergent constraints provided gives a hint to which kind of emergent constraints one can look out for in an ensemble of high-dimensional Global Climate Models (GCMs). To find an emergent constraint for climate sensitivity by data mining in a CMIP5 ensemble proved fruitless (Caldwell et al., 2014). Using the susceptibilities to find new emergent

constraints does not seem to have a direct advantage above directly looking for plausible correlations, but susceptibilities might provide additional information. For example, when a susceptibility shows a resonance at a certain frequency over the ensemble of models, this could suggest that the same feedback is present in all simulations.

In a high-dimensional dynamical system eigenfunctions and eigenvalues can be accessed with the help of transfer operators, associated with the propagation of probability densities associated with the Fokker-Planck operator. The eigenfunctions that

lie on the invariant measure are then computed by making use of the ergodic properties of the climate system. To overcome the burden of high-dimensionality, a reduced transfer operator can be computed from a very long simulation, from which the eigenfunctions on the attractor are approximated (Tantet, 2016). However, a forcing on the system does not generally lie only on the attractor and should be split into a part parallel and perpendicular to the attractor. Consequently, the igenvectors off the attractor cannot a priori be ignored (Lucarini and Sarno, 2011). Gritsun and Lucarini (2017) showed that indeed for some

geophysical systems, specifically quasi-geostrophic flow with orographic forcing, the response to the forcing may have no resemblance to the unforced variability in the same range of spatial and temporal scales.

In conclusion, while the current theoretical framework provides an understanding on how emergent constraints may arise in low-dimensional stochastic dynamical systems, its application to output from GCMs, in particular in finding novel and useful emergent constraints, is a challenging issue for future work.

*Acknowledgements.* We thank Frank Lunkeit for his help with the PlaSim simulations and Alexis Tantet is thanked for useful discussions. Both authors acknowledge support by the Netherlands Earth System Science Centre (NESSC), financially supported by the Ministry of Education, Culture and Science (OCW), Grant no. 024.002.001



**Appendix: Response function expansion**

For $A = x$, we find from (2.5) that

$$R_A(t) = \int_{-\infty}^{\infty} x\, e^{\mathscr{L}^* t}\left(-\frac{\partial \bar{p}_e}{\partial x}\right) dx. \tag{5.1}$$

Using the expression for the equilibrium solution $\bar{p}_e$, we find

$$-\frac{\partial \bar{p}_e}{\partial x} = \frac{2}{\sigma} V'(x)\bar{p}_e \tag{5.2}$$

and hence (5.1) becomes

$$R_A(t) = \int_{-\infty}^{\infty} x\, e^{\mathscr{L}^* t}\left(\frac{2}{\sigma} V'(x)\bar{p}_e\right) dx. \tag{5.3}$$

With the standard $L^2$-inner product, the adjoint of $\mathscr{L}$ determined as $\langle \mathscr{L}^* g, h\rangle = \langle g, \mathscr{L} h\rangle$, where $\mathscr{L}$ is the generator of the OU process, is given by

$$\mathscr{L}u = V'(x)\frac{\partial u}{\partial x} + \frac{\sigma}{2}\frac{\partial^2 u}{\partial x^2} \tag{5.4}$$

Using this property in (5.3), we find

$$\langle x, e^{\mathscr{L}^* t}(V'(x)\bar{p}_e)\rangle = \langle e^{\mathscr{L} t} x, V'(x)\bar{p}_e\rangle \tag{5.5}$$

and hence

$$R_A(t) = \frac{2}{\sigma}\int_{-\infty}^{\infty} e^{\mathscr{L} t}(x)\, V'(x)\bar{p}_e\, dx \tag{5.6}$$

Next an inner product $\langle g, h\rangle_{\bar{p}_e}$ is defined as

$$\langle g, h\rangle_{\bar{p}_e} = \int_{-\infty}^{\infty} gh\bar{p}_e\, dx \tag{5.7}$$

As a next step, let $\lambda_l$ and $\phi_l$ be the eigenvalues of the generator, i.e. solutions $v$ of

$$\mathscr{L}\phi = -\lambda\phi \tag{5.8}$$



For reversible processes, these eigenvalues are real, positive and discrete under the inner product $\langle,\rangle_{\bar{p}_e}$. The eigenfunctions form a complete orthonormal basis, such that $\langle\phi_n,\phi_m\rangle_{\bar{p}_e}=\delta_{nm}$ (Pavliotis, 2014). Now $e^{\mathscr{L}t}(x)$ represents solutions $u(x,t)$ of the problem

$$\frac{\partial u}{\partial t}=\mathscr{L}u \qquad (5.9)$$

with initial condition $u(x,0)=x$. We can expand $u$ into eigenfunctions as

$$u(x,t)=\sum_{l=1}^{\infty}\alpha_l\phi_l(x)e^{-\lambda_l t} \qquad (5.10)$$

From the initial condition, we find

$$\sum_{l=1}^{\infty}\alpha_l\phi_l(x)=x \qquad (5.11)$$

and using the orthogonality of the $\phi_l$ under the inner product $\langle,\rangle_{\bar{p}_e}$, we find

$$\alpha_l=\langle x,\phi_l\rangle_{\bar{p}_e} \qquad (5.12)$$

On the other hand, substituting the expression for $u$ into (5.6) gives

$$\int_{-\infty}^{\infty}\sum_{l=1}^{\infty}\alpha_l\phi_l(x)e^{-\lambda_l t}V'(x)\bar{p}_e\,dx=\sum_{l=1}^{\infty}\beta_l e^{-\lambda_l t} \qquad (5.13)$$

where

$$\beta_l=\alpha_l\langle V'(x),\phi_l\rangle_{\bar{p}_e}=\langle x,\phi_l\rangle_{\bar{p}_e}\langle V'(x),\phi_l\rangle_{\bar{p}_e} \qquad (5.14)$$

Repeating the derivation with a general observable $A=f(x)$ gives $\langle f(x),\phi_l\rangle_{\bar{p}_e}\langle V'(x),\phi_l\rangle_{\bar{p}_e}$. The first term in $\beta_l$ denotes the projection of the observable on the eigenfunctions and could intuitively be interpreted (for $l>0$) as the amenability of the observable to change. The second projection term in $\beta_l$ can be understood to be the amenability of the whole system to change under the influence of the forcing field.



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




| Reference | Climate predictor | Future climate predictand | Type |
|---|---|---|---|
| Knutti et al. (2006) | Seasonal cycle land temperature amplitude | ECS | DD |
| Hall and Qu (2006); Qu and Hall (2014) | Springtime SAF | SAF under climate warming | DD |
| Boe et al. (2009) | Arctic sea ice extent trend 1979-2007 | Arctic sea ice extent | DD |
| Clement et al. (2009) | Sensitivity LLC to pacific decal variability | Sign LLC feedback | DD |
| Trenberth and Fasullo (2010) | SH net radiation TOA | ECS | IS |
| Fasullo and Trenberth (2012) | Mid-tropospheric RH over ocean in subsidence region | ECS | IS |
| Bracegirdle and Stephenson (2013) | Arctic SAT | Arctic SAT under climate warming | DS |
| Gordon and Klein (2014) | Sensitivity of extra-tropical LLC optical depth to temperature | Extra-tropical LLC optical depth response to climate warming. | DD |
| Qu et al. (2014) | Sensitivity of LLC cover to SST | LCC cover changes under climate warming | DD |
| Sherwood et al. (2014) | Strength cloud-scale and large-scale lower tropospheric mixing over oceans | ECS | IS |
| Su et al. (2014) | RH & cloud fraction tropics | ECS | IS |
| Wenzel et al. (2014) | Short-term sensitivity of atmospheric carbon dioxide | Sensitivity tropical land carbon storage to climate warming | DD |
| Tian (2015) | Precipitation & mid-tropospheric RH asymmetry bias (for ITCZ) | ECS | IS |
| Kwiatkowski et al. (2017) | Tropical primary production under ENSO-driven SST variations | Tropical primary production under climate change | DD |
| Cox et al. (2018) | Function of autocorrelation of GMST | ECS | ID |

**Table 1.** Application of our classification of emergent constraints to a selection of examples found in literature. DD is a direct dynamical constraint, DS a direct static constraint and IS is an indirect static constraint, while ID denotes indirect dynamical emergent constraints. Abbreviations stand for RH: relative humidity, ITCZ: inter-tropical convergence zone, TOA: top of atmosphere, SH: southern hemisphere, ECS: equilibrium climate sensitivity, LLC: low-level cloud, SAF: snow-albedo feedback. SAT: surface air temperature. The emergent constraint found by Trenberth and Fasullo (2010) seems to be spurious: no physical mechanism was proposed and it did not appear in different ensembles, such as CMIP5 (Grise et al., 2015).



| Constant | Value | | Constant | Value |
|----------|-------|---|----------|-------|
| $c_T$ | $5.0 \times 10^8 \text{ J/m}^2/\text{K}$ | | $\epsilon$ | 1.0 |
| $A$ | $20.5 \text{ W/m}^2$ | | $\sigma_B$ | $5.67 \times 10^{-8} \text{ W/m}^2/\text{K}^4$ |
| $Q_0$ | $342 \text{ W/m}^2$ | | $\alpha_{min}$ | 0.2 |
| $Q_s$ | $115 \text{ W/m}^2$ | | $\alpha_{amp}$ | 0.05–0.5 |
| $G$ | $150 \text{ W/m}^2$ | | $k$ | 0.5 |
| $C_{ref}$ | 280 ppmv | | $T_h$ | 284 K |
| $\tau_s$ | $4.0 \times 10^6 \text{ s}$ | | $\sigma_T$ | $1.0 \times 10^{-7} \text{ K}^2/\text{s}$ |
| $\sigma_\alpha$ | $1.0 \times 10^{-5} \text{ s}^{-1}$ | | | |

**Table 2.** Constants for the energy balance model.





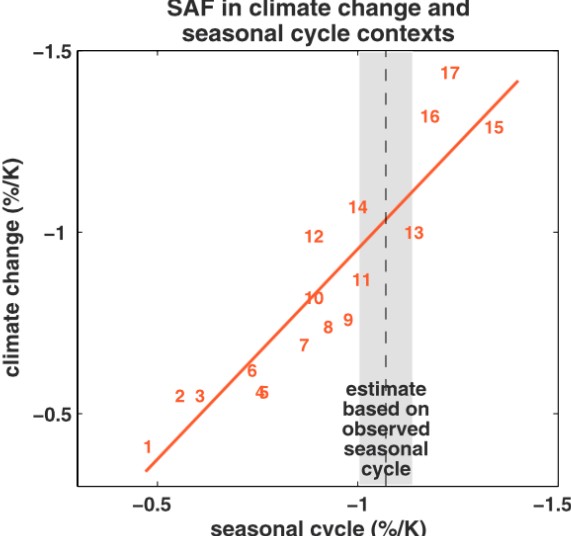

**Figure 1.** The emergent constraint on snow-albedo feedback $\frac{\Delta\alpha_s}{\Delta T_s}$ (from Hall and Qu (2006), $\alpha_s$ given in units of %). This is an example of a direct emergent constraint (it links the SAF in both past and future time) and a dynamical emergent constraint (it uses a response to a seasonal forcing as its predictor).




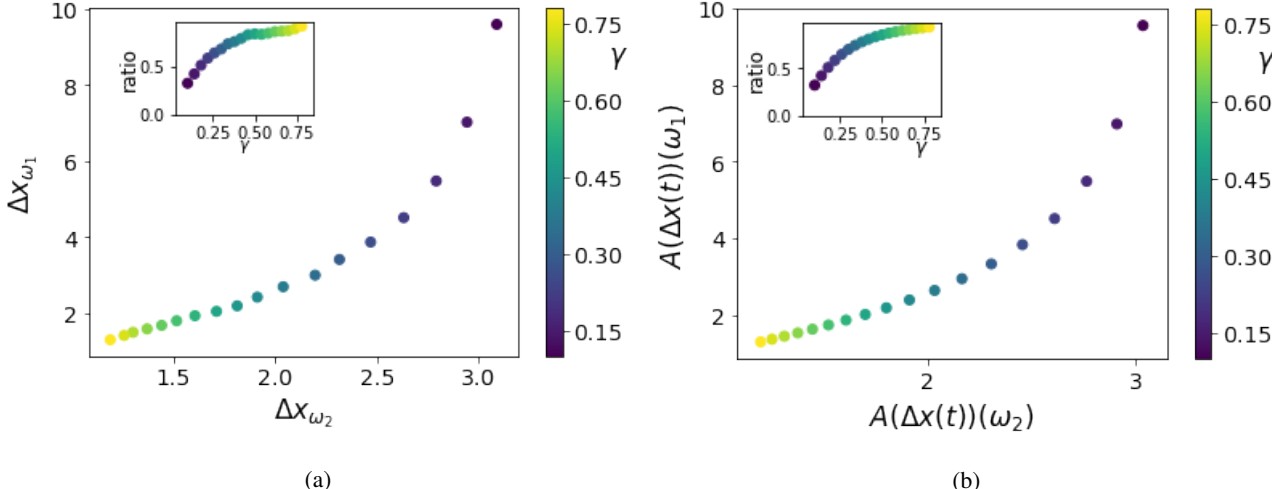

|          |          |
|----------|----------|
| (a)      | (b)      |

**Figure 2.** *(a)* Response to forcings at two different frequencies of the one-dimensional Ornstein-Uhlenbeck process. Shown is the average of a 500-member simulation of trajectories *(b)* The susceptibility at these frequencies, whose ratio is given in the inset figure. This is an example of a direct dynamical emergent relationship.



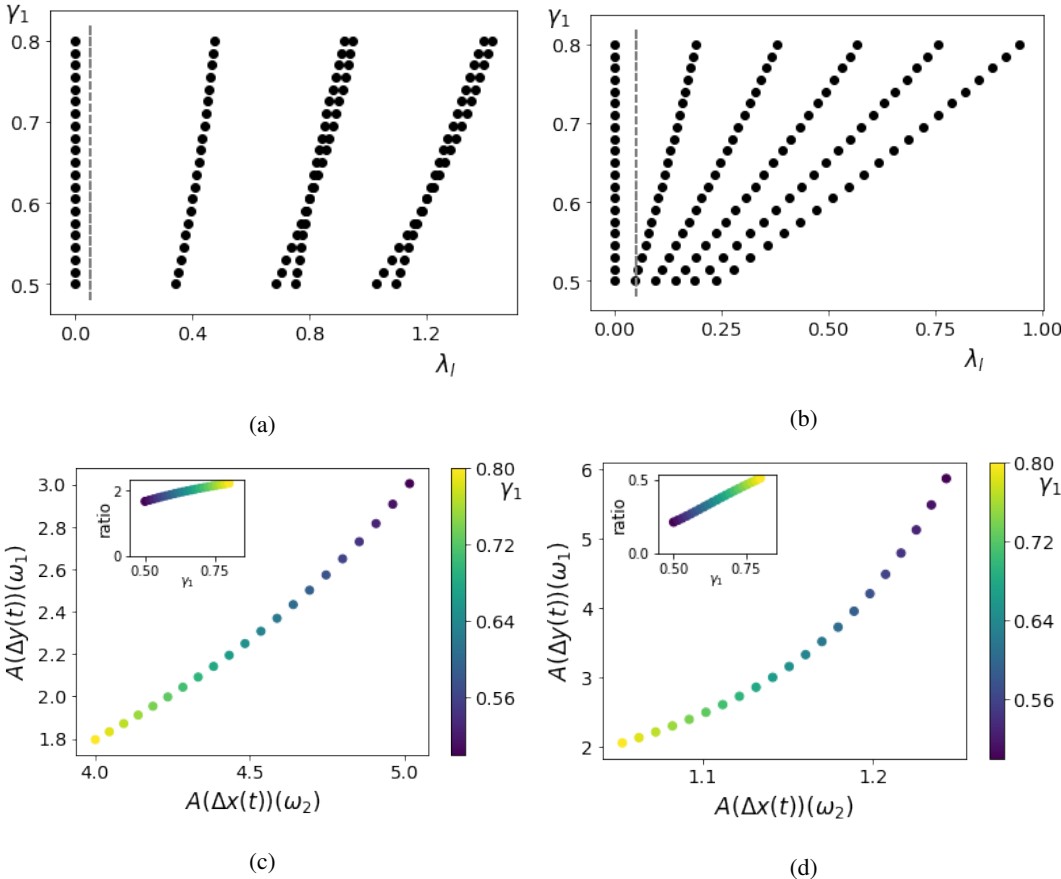

**Figure 3.** Eigenvalue spectrum for *(a)* $\delta = 0.2$ and *(b)* $\delta = 0.5$. The dashed line corresponds to the frequency $\omega_2$ of the fast forcing *(c,d)* Corresponding susceptibilities, with their ratio in the inset figures. This is an example of an indirect dynamical emergent relationship. Note that for reasons of numerical stability, the range of $\gamma_1$ is different than that of $\gamma$ in figure 2.





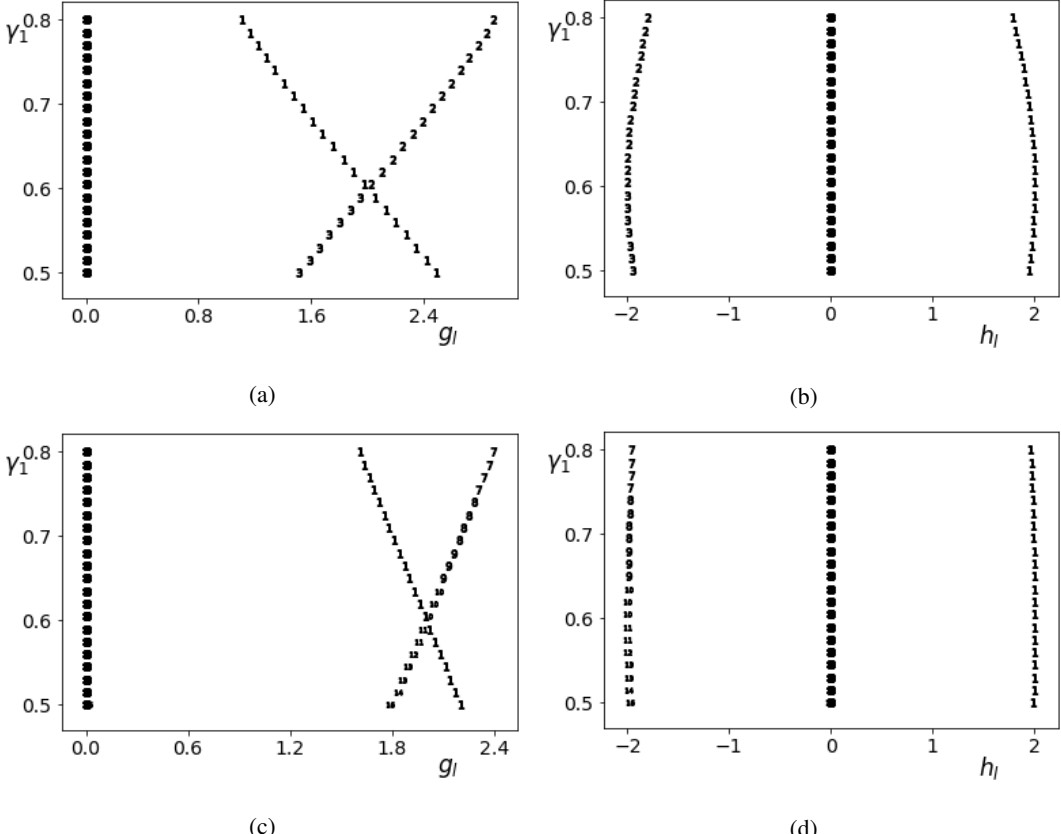

**Figure 4.** *(a,b)* Projections $g_l$ (of predictor variable) and $h_l$ (of predictand variable) for a weakly coupled two-dimensional OU system with $\delta = 0.2$, *(c,d)* same for $\delta = 0.5$.


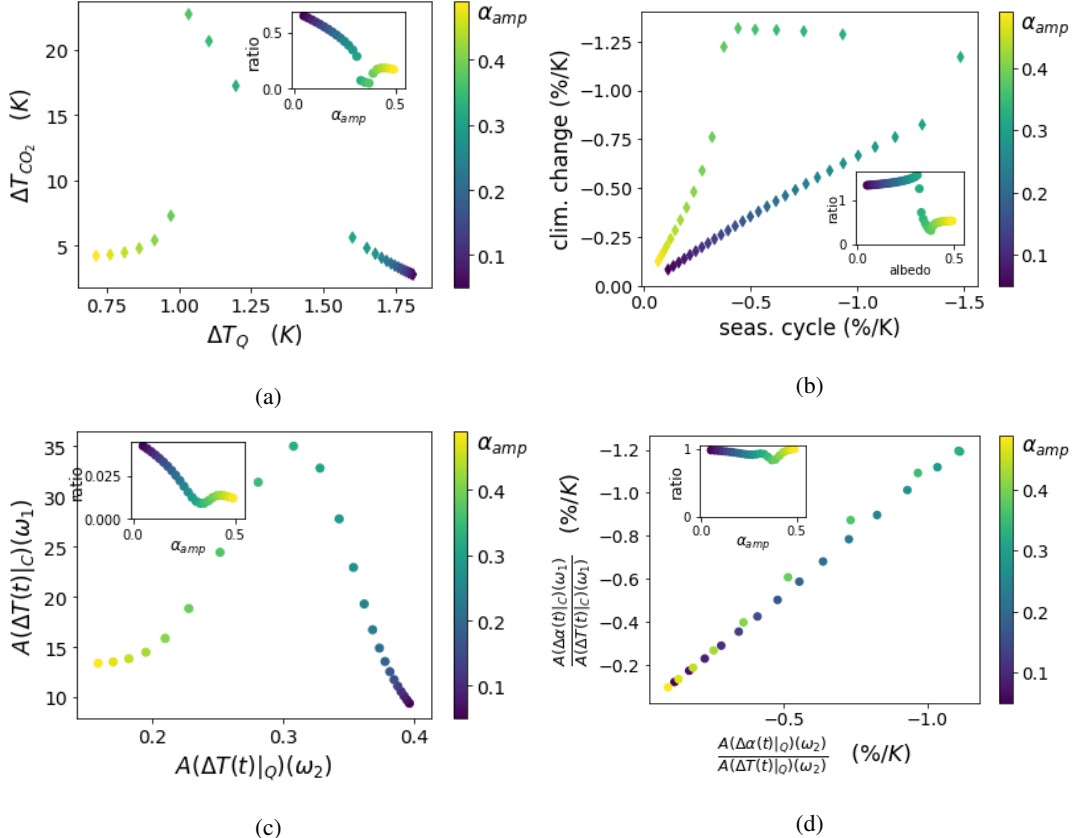

**Figure 5.** *(a)* The relation between temperature response to the seasonal cycle and the temperature response to greenhouse gas forcing. *(b)* The strength of the snow-albedo feedback to solar and greenhouse gas forcing on different time scales. In the inset: their ratio as a function of $\alpha_{max}$. For clearness, *(a,b)* are shown without noise. *(c)* The susceptibilities for temperature as the observable *(d)* The ratio of albedo and temperature susceptibilities and their ratio (RFS).





**Figure 6.** *(a)* Eigenvalues of the EBM depending on the amplitude of the albedo function for the simple EBM. The zero eigenvalues correspond to the invariant measure. *(b)* The extended EBM. *(c)* Albedo projection terms for solar forcing ($\alpha_l$) as defined in Equation 4.7 where the markers denote $l$ *(d)* Same for temperature $\beta_l$ *(e,f)* projections terms for GHG forcing $\gamma_l$ and $\delta_l$ respectively.





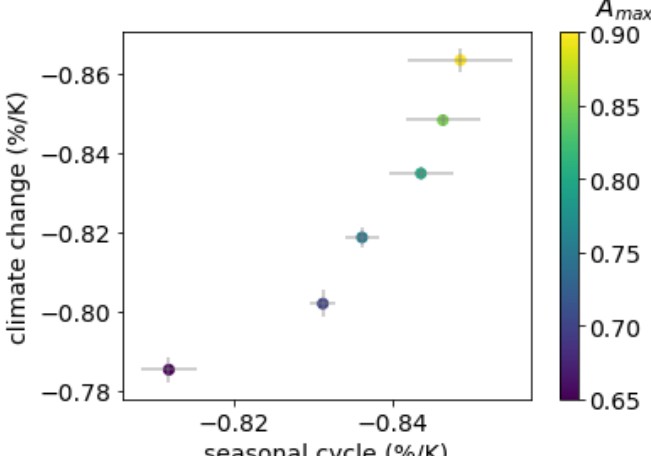

**Figure 7.** Same as figure 1, but now results from *PlaSim*