# Peer review of "A mathematical approach to understanding emergent constraints"

_Earth System Dynamics, 2018_

## Referee Comment (RC1) · Anonymous Referee #1 · 25 May 2018

This manuscript is dealing with the problem of the understanding of emergent constraints in projections based on climate models. The main idea is to develop a mathematical framework based on the linear response theory. The approach is applied in the context of several models of increasing complexity. A classification of emergent constraints is also proposed. This is a very interesting approach to the problem that is worth publishing.

The organization of the manuscript is however confusing to me. Section 2 is mixing the general development of the approach and the application to an Ornstein-Uhlenbeck process. It is therefore difficult to figure out what is general or not. I would suggest the authors to reorganize this section 2 (and also section 3), by first presenting the general framework based on Response theory and then the specific application to the O-U

process, maybe by putting a section 2.1 and a second section 2.2 (or by rearranging section 2 and 3 together).

Specific points:

- Page 1, Line 14, remove "variable"

- Page 3, Line 12. O=x. Is it really a mean value?

- Page 4, Eq 2.12-2.13. The way to compute the g_l and h_l should be explained.

- Page 5, Eq 3.1. Is it only valid for O-U process? This point is related to the general comment above. What is general and what is specific to the O-U process? This should be clarified.

- Page 6. Same as the previous point for Eq 3.4 and 3.5

- Page 6, line 29. Remove "the".

- Page 8, Eq. 4.3. References to these type of models are needed. You can go back to the pioneers on that topic.

- Page 9. Eq. 4.6, one omega_2 should be omega_1, I guess.

- Page 9. Eq 4.8. Is this model presented elsewhere? References are needed.

- Page 10. Line 12. What means "a 50-year spin-up was used"? Before the 20th and 21th centuries?

---

## Referee Comment (RC2) · V. Lucarini (Referee) · 22 Jun 2018

Dear authors,

your contribution is extremely relevant and I supports its publication pending the minor revisions I suggest in the annotated .pdf file.

This contribution is relevant because it attempts at giving a systematic treatment of the concept of emergent constraints, which had been - up to now - pretty vague. This lack of clarity needs to to be addressed, and this paper does exactly this. The methodology used here - essentially, response theory - is a powerful mathematical idea that I also find extremely relevant.

I see as main (yet minor) pitfalls of your paper the somewhat confused presentation in

[Figure]

Section 2 and the fact that the classification of the emergent constraints is also a bit unclear.

I think you should also discuss a bit more in detail the difference between considering multiple models, instead of one with parametric modulations.

Additionally, you might find useful a recent preprint of mine:

https://arxiv.org/abs/1806.03983

where I address the problem of looking at observables as predictands and predictors. This problem is (briefly) mentioned in your paper, but maybe my preprint can be useful for discussing your results.

Overall, this paper is a very good piece of work.

Valerio Lucarini

Please also note the supplement to this comment:
https://www.earth-syst-dynam-discuss.net/esd-2018-15/esd-2018-15-RC2-supplement.pdf

**Supplement:**

[revised manuscript text omitted]

(a)                                                                                              (b)

**Figure 2.** *(a)* Response to forcings at two different frequencies of the one-dimensional Ornstein-Uhlenbeck process. Shown is the average of a 500-member simulation of trajectories *(b)* The susceptibility at these frequencies, whose ratio is given in the inset figure. This is an example of a direct dynamical emergent relationship.

[Figure]

**Figure 3.** Eigenvalue spectrum for *(a)* $\delta = 0.2$ and *(b)* $\delta = 0.5$. The dashed line corresponds to the frequency $\omega_2$ of the fast forcing *(c,d)* Corresponding susceptibilities, with their ratio in the inset figures. This is an example of an indirect dynamical emergent relationship. Note that for reasons of numerical stability, the range of $\gamma_1$ is different than that of $\gamma$ in figure 2.

[Figure]

**Figure 4.** *(a,b)* Projections $g_l$ (of predictor variable) and $h_l$ (of predictand variable) for a weakly coupled two-dimensional OU system with $\delta = 0.2$, *(c,d)* same for $\delta = 0.5$.

[Figure]

**Figure 5.** *(a)* The relation between temperature response to the seasonal cycle and the temperature response to greenhouse gas forcing. *(b)* The strength of the snow-albedo feedback to solar and greenhouse gas forcing on different time scales. In the inset: their ratio as a function of $\alpha_{max}$. For clearness, *(a,b)* are shown without noise. *(c)* The susceptibilities for temperature as the observable *(d)* The ratio of albedo and temperature susceptibilities and their ratio (RFS).

[Figure]

**Figure 6.** *(a)* Eigenvalues of the EBM depending on the amplitude of the albedo function for the simple EBM. The zero eigenvalues correspond to the invariant measure. *(b)* The extended EBM. *(c)* Albedo projection terms for solar forcing ($\alpha_l$) as defined in Equation 4.8 where the markers denote $l$ *(d)* Same for temperature $\beta_l$ *(e,f)* projections terms for GHG forcing $\gamma_l$ and $\delta_l$ respectively.

[Figure]

**Figure 7.** Same as figure 1, but now results from *PlaSim*

---

## Author Comment (AC1) · 7 Jul 2018

We thank the referee for the careful reading and the useful comments and will adapt the manuscript accordingly. Below is a point by point reply with the referee's comments first, followed by our reply and the changes in manuscript.

1. Comment of the referee: This manuscript is dealing with the problem of the understanding of emergent constraints in projections based on climate models. The main idea is to develop a mathematical framework based on the linear response theory. The approach is applied in the context of several models of increasing complexity. A classification of emergent constraints is also proposed. This is a very interesting approach to the problem that is worth publishing. The organization of the manuscript is however

confusing to me. Section 2 is mixing the general development of the approach and the application to an Ornstein-Uhlenbeck process. It is therefore difficult to figure out what is general or not. I would suggest the authors to reorganize this section 2 (and also section 3), by first presenting the general framework based on Response theory and then the specific application to the O-U C1 ESDD Interactive comment Printer-friendly version Discussion paper process, maybe by putting a section 2.1 and a second section 2.2 (or by rearranging section 2 and 3 together).

Author's response: We will follow the suggestion of the referee to clarify better what is general and what applies to the OU case and reorganize the paper accordingly.

Changes in the text: The sections 2 and 3 will be reorganized to better separate the general and specific cases.

2. Comment of the referee: Page 1, Line 14, remove "variable"

Author's response: Suggestion followed.

Changes in the text: 'variable' will be removed.

3. Comment of the referee: Page 3, Line 12. O=x. Is it really a mean value?

Author's response: It was meant to indicate the identity operation.

Changes in the text: We will mention this now in words in the revised text.

4. Comment of the referee: Page 4, Eq 2.12-2.13. The way to compute the $g\_l$ and $h\_l$ should be explained.

Author's response: Suggestion followed.

Changes in the text: A reference to the appendix will be added. The appendix will contain the explicit computation of $g\_l$ and $h\_l$.

5. Comment of the referee: Page 5, Eq 3.1. Is it only valid for O-U process? This point is related to the general comment above. What is general and what is specific to the

O-U process? This should be clarified.

Author's response: This is valid in general.

Changes in the text: In the revised paper this will be made clear by restructuring the material as mentioned under comment 1 above.

6. Comment of the referee: Page 6. Same as the previous point for Eq 3.4 and 3.5

Author's response: These results are also general.

Changes in the text: In the revised paper this will be made clear by restructuring the material as mentioned under comment 1 above.

7. Comment of the referee: Page 6, line 29. Remove "the".

Author's response: Suggestion followed.

Changes in the text: 'the' will be removed.

8. Comment of the referee: Page 8, Eq. 4.3. References to these type of models are needed. You can go back to the pioneers on that topic.

Author's response: Suggestion followed.

Changes in the text: Several references will be added, e.g., Budyko (1969), Sellers (1969), Fraedrich (1979) and Satura (1981) .

9. Comment of the referee: Page 9. Eq. 4.6, one omega_2 should be omega_1, I guess.

Author's response: Thanks.

Changes in the text: The equation (4.6) will be corrected.

10. Comment of the referee: Page 9. Eq 4.8. Is this model presented elsewhere? References are needed.

Author's response: No, as far as we know this is the first time such a formulation has been used.

Changes in the text: None.

11. Comment of the referee: Page 10. Line 12. What means "a 50-year spin-up was used"? Before the 20th and 21th centuries?

Author's response: A spin-up was used before the 20th century, so from 1850-1900. This will be clarified in the revised text.

---

## Author Comment (AC2) · 7 Jul 2018

We thank the referee for the careful reading and the useful comments and will adapt the manuscript accordingly. Below is a point by point reply with the referee's comments first, followed by our reply and the changes in manuscript.

1. Comment of the referee: I see as main (yet minor) pitfalls of your paper the somewhat confused presentation in Section 2 and the fact that the classification of the emergent constraints is also a bit unclear.

Authors' response: We will aim to clarify both issues.

Changes in the text: Section 2 and 3 will be rewritten as to separate the more general derivation from the Ornstein-Uhlenbeck example. The classification of the emergent

constraints in section 3 will be made more clear as per the suggestions of the referee.

2. Comment of the referee: I think you should also discuss a bit more in detail the difference between considering multiple models, instead of one with parametric modulations.

Author's response: Suggestion followed.

Changes in the text: We will address the differences between the two cases (multiple models vs parameter variation in a single model) in the revised discussion section of the paper.

3. Comment of the referee: Additionally, you might find useful a recent preprint of mine: https://arxiv.org/abs/1806.03983 where I address the problem of looking at observables as predictands and predictors. This problem is (briefly) mentioned in your paper, but maybe my preprint can be useful for discussing your results.

Authors' response: We have read the paper and the material is indeed highly interesting and relevant for our paper.

Changes in the text: It will be used when rewriting part of section 3 on the classification of the emergent constraints.

4. Comment of the referee: Please also note the supplement to this comment: https://www.earth-syst-dynam-discuss.net/esd-2018-15/esd-2018-15-RC2- supplement.pdf

Authors' response: Many thanks for the very useful questions, remarks and suggestions for changes; they are discussed below.

Changes in the text: All these suggestions will be taken into account in the revised paper as per the point-by-point reply below.

5. Comment of the referee: Section 2: I think you should frame response theory in general terms, and then propose this (relevant and illustrative) example. Otherwise,
the reader can be a bit confused.

Authors' response: Good point. Now we only refer to the general theory of response theory in the introduction and discussion, but this is the most logical place.

Changes in the text: Section 2 will be rewritten to better separate the general results from that of the example.

6. Comment of the referee: I also think you should cite the Hairer and Majda 2010 paper regarding response theory for stochastic systems and the original Ruelle 1998, 2009 papers on response theory in a deterministic setting.

Authors' response: We indeed mainly referred to the book by Pavliotis, but it is indeed good to cite the papers in which these advancements were made.

Changes in the text: References to the papers by Ruelle (1998, 2009) and Hairer and Majda (2010) will be added.

7. Comment of the referee: Page 3, line 30 (in the version attached to review, which is the 2-column version of the ESDD paper): be more specific. Firstly, an emergent constraint can be either direct or indirect. In the direct case, the predictor and predictand are the same observable, while in the indirect case the predictor variable and predictand variable have to be linked via a physical process.

Authors' response: Indeed, it was not made explicit that indirect simply means not the same variable.

Changes in the text: The sentence will be changed to: Firstly, an emergent constraint can be either direct or indirect. In the direct case, the predictor and predictand are the same observable, while in they are not. In the latter case, the predictor variable and predictand variable have to be closely linked, for instance via a physical process.

8. Comment of the referee: Page 3, line 69: Do you mean: when taking ensemble averages?

Authors' response: Not when we talk about an ensemble generated by varying a parameter. For a linear relationship to appear (A = Cst B), the ratio of susceptibilities should be constant (A/B= Cst).

Changes in the text: We will write out that (A= Cst B) to show readers nothing special is going on here.

9. Comment of the referee: Page 3, line 79: The equation 3.2 is unclear

Authors' response: Indeed, the accompanying text was quite unclear.

Changes in the text: We will add: "One variable (B) can act as forcing to a second variable (O), while being itself forced by some external variable (F). Furthermore, often the forcing patterns are not exactly the same for the short and long periodic forcing, leading to:"

10. Comment of the referee: Page 4, line 7 Yes, but you have lost sensitivity on \gamma

Authors' response: This is the point. If there was still a dependence on gamma, no linear relationship would have appeared.

Changes in the text: By making equation 3.1 more clear, we will make this more apparent.

11. Comment of the referee: Page 4, line 18: This has to do in fact, with the sum rules and asymptotic properties studied in Lucarini and Sarno (2011).

Authors' response: Thanks for this remark, but it would be too much detailed to discuss that here.

Changes in the text: None.

12. Comment of the referee: Page 4, line 55: isn't this just a special case of 3.1, just with \omega_2=0?
Authors' response: This is not a special case of equation 3.1. Here we are not interested in the response to forcing at all, only in the expectation value. This was explained in the text in the left column, but apparently not sufficiently clear.

Changes in the text: The difference between the low frequency limit $\omega_2 \to 0$ of (3.1) and equation (3.5) will be explained in the revised text.

13. Comment of the referee: Page 5, line 31. maybe it can be made a bit clearer?

Authors' response: The reason why was indeed omitted.

Changes in the text: The last sentence of this paragraph will be changed to clarify the role of the signal-to-noise ratio In finding an emergent constraint.

14. Comment of the referee: Page 7, line 15: As long as different models are closely related and structural differences can be approximately parameterized.

Authors' response: Agreed.

Changes in the text: This will be added to the revised text at the end of section 4 and also when discussing the cases (multiple models vs parameter variation in a single model) in the revised section 5.

15. Comment of the referee: Page 7, line 29: I believe that this specific classification should be better explained.

Authors' response: Agreed.

Changes in the text: This has been now made clearer in section 3 and will be repeated in the revised discussion section.